# Polymer Free Amphilimus Drug Eluting Stent for Infrapopliteal Arterial Disease in Patients with Critical Limb Ischemia: A New Device in the Armamentarium

**DOI:** 10.3390/medicina59010039

**Published:** 2022-12-24

**Authors:** Konstantinos Tigkiropoulos, Manolis Abatzis-Papadopoulos, Katerina Sidiropoulou, Kyriakos Stavridis, Dimitrios Karamanos, Ioannis Lazaridis, Nikolaos Saratzis

**Affiliations:** Division of Vascular Surgery, 1st Surgical Department, Faculty of Health Sciences, Aristotle University, Papageorgiou General Hospital, 56429 Thessaloniki, Greece

**Keywords:** critical limb ischemia, below the knee, drug eluting stents, Cre8, polymer free, endovascular

## Abstract

***Background and Objectives*:** Endovascular technologies have significantly improved the outcome of patients with critical limb ischemia (CLI). Drug eluting stents (DES) have documented their efficacy against percutaneous transluminal angioplasty (PTA) and bare metal stents (BMS) in infrapopliteal arterial occlusive disease. However, late in-stent neoatherosclerosis may lead to vascular lumen loss and eventually thrombosis. Polymer free DES constitute a new technology aiming to improve long term patency which their action is still under investigation. The purpose of this study is to report the mechanism of action and to provide a literature review of a novel polymer free amphilimus eluting stent (Cre8, Alvimedica, Instabul, Turkey) in infrapopliteal arterial disease. ***Methods*:** Publications listed in electronic databases, European Union Drug Regulating Authorities Clinical Trials Database, as well as scientific programmes of recent interventional vascular conferences were searched. Three studies were included. We analyzed primary and secondary patency, major amputation rate, freedom from CD-TLR, and mortality. ***Results*:** Cre8 was implanted in 79 patients with CLI. Most of the patients (*n* = 65) were Rutherford class 5–6 (82.3%), and diabetes mellitus (DM) was present in 66 patients (83.5%). Mean primary patency was 82.5% at 12 months. Mean lesion stented length was 20 mm and 35 mm in two studies. Mean limb salvage was 91.3% at 12 months. Freedom from CD-TLR was reported in two out of the three studies and was 96% and 83.8%. Mortality was 15% and 23.8% in the same studies, whilst it was not reported in one study. ***Conclusion*:** Stenting of infrapopliteal arteries with Cre8 is safe and feasible in patients with CLI and diabetes. All studies have shown very good primary patency and freedom from CD-TLR at 12 and 24 months. Larger observational prospective studies and randomized trials are necessary to establish long term effectiveness and clinical outcomes using the non-polymer Cre8 DES.

## 1. Introduction

Critical limb ischemia is defined as chronic rest pain with/or the presence of tissue loss due to decreased perfusion of lower extremities. It is the most severe form of peripheral arterial disease (PAD) with increasing prevalence [1]. It is associated with limb and life-threatening pathology and is considered an important health problem with social and financial consequences [2]. CLI patients suffer from serious comorbidities, such as diabetes and chronic renal disease in high prevalence [3,4,5], presenting with multilevel steno-occlusive disease, with infrapopliteal arteries having long atheromatous lesions, and with high calcified burden, negatively affecting long term arterial patency. Diabetic patients with CLI have increased risk of lower extremity amputation and increased mortality rate exceeding 50% at five years [6].

Endovascular therapy with percutaneous transluminal angioplasty (PTA) and “bailout” stenting is considered as first line treatment for CLI patients [7,8]. However, it is associated with decreased primary patency rate and may need secondary interventions to reestablish blood flow to BTK arteries [9]. Drug eluting technology was primarily investigated and applied in coronary arterial disease (CAD). Due to similarities in vessel diameter with tibial arteries, DES have found fertile ground in CLI patients with infrapopliteal atherosclerotic disease with small and medium length lesions. Their main function apart from overcome flow limiting dissection and elastic recoil, is to inhibit neointimal hyperplasia restenosis by the antiproliferative agent loaded in the scaffold. Several randomized control trials (RCTs) have been performed the last 15 years, demonstrating the superiority of DES technology against PTA and bare metal stents (BMS) in primary patency rate, target lesion revascularization (TLR), and major amputation [10,11,12,13,14,15]. A meta-analysis by Fusaro et al. demonstrated a decreased rate of amputations and re-interventions of patients with infrapopliteal DES over PTA and BMS [16].

However, the antiproliferative action of the drugs may prolong arterial wall healing and promote late stent thrombosis. Additionally, polymer coated scaffolds, which act as a barrier between arterial wall and stents, remain in contact with vascular endothelium after the release of the antiproliferative agent contributing to an inflammatory reaction, neointimal hyperplasia, restenosis, and in-stent thrombosis. The technological evolution of DES is progressed over years, using more active antirestenotic agents with improved pharmaco-kinetic properties and polymer free or biodegradable polymer stents to eliminate late adverse inflammatory interaction between scaffold and arterial wall. Everolimus and Sirolimus eluting stents have been studied in previous RCTs [13,14]. Novel limus agents have been produced by manufacturers, e.g., zotarolimus and biolimus, and have been loaded in special scaffolds [17,18]. The aim of this study is to present a new polymer free amphilimus eluting stent (Cre8, Alvimedica, Instanbul, Turkey) for infrapopliteal arterial lesions, which was initially developed for coronary arterial disease (CAD). The mechanism of action will be analyzed, and we will provide the most recent clinical data of its application in CLI patients.

## 2. Mechanism of Polymer-DES Restenosis

DES technology was developed to decrease the adverse events occurred in percutaneous coronary interventions (PCI) with PTA and BMS deployment [19]. Rapamycin analogues exert an immunosuppressant, anti-proliferative, and anti-inflammatory action and become the pharmaceutical agent of choice for DES [20]. Polymer coating was developed on the metallic scaffold to ensure controlled release of the drug with appropriate eluting pharmaco-kinetics, do not interact with the anti-proliferative agent and to be biocompatible [21]. However, first generation DES were associated with high risk of late stent thrombosis. Autopsy studies have shown, apart from poor stent strut coverage, that polymer persistence may induce a chronic vessel inflammation, delayed arterial healing, and neo-atherosclerosis, resulting in late in stent lumen loss and eventually thrombosis [22,23]. Additionally local hypersensitivity reaction as defined by the presence of eosinophils secondary to polymer is considered as risk factor as well as intrusion of stent struts to the atherosclerotic necrotic core at bifurcations [24,25]. Denardo et al. showed that polymer surface DES before and after balloon expansion demonstrate severe irregularities and fractures [24]. Joner et al. concluded that the persistence of fibrin and incomplete endothelialization beyond one month from the deployment of the polymer coated DES is the main pathological substrate for late stent thrombosis [25]. Non-polymer-related risk factors for DES restenosis are stent under expansion due to calcified lesions and elastic recoil, stent fracture with local trauma at the arterial wall preserved by vessel movement during cardiac cycle, and over dilatation of an under-sized stent and non-uniform drug deposition [26].

## 3. Device Characteristics

In an attempt to eliminate the adverse effects of polymer coating, polymer free DES were developed. One of the novel DES that has been introduced as a coronary stent in the European market in 2011 is the Cre8 amphilimus eluting stent (Alvimedica, Instabul, Turkey). Cre8 DES uses an optimized L605 Cobalt-Chromium (CoCr) alloy platform to provide precise positioning and maximum radial force. The strut thickness ranges from 70 to 80 μm, stent diameters range from 2.25 to 4.50 mm and stent lengths from 8 to 46 mm (Table 1). The homogeneous scaffolding of the stent allows to maintain its mechanical properties without any change after dilatation even in the presence of arterial bifurcations without compromising arterial flow (Figure 1). The presence of midpoint connections among the cells permits no foreshortening of the device during expansion.

The device has 2 platinum radiopaque markers at both stent ends enabling precise deployment in tibial arteries. One the unique characteristics of Cre8 Des is the coating of the strut surface. It consists of a bio inducer surface (BIS), which is a 2nd generation ultra-thin pure carbon coating, less than 0.3 μm in size, that improves biocompatibility and hemocompatibility [27]. It is applied in the platform before drug loading. BIS is the only interface with endothelial layer and arterial blood flow promoting faster endothelization and healing as well as acting as an antithrombotic coating (Figure 2) [28]. It does not permit direct contact of the scaffold to the artery acting as a barrier to metal ions release, e.g., Nickel, when used on metal alloy avoiding allergic reaction. BIS does not affect the mechanical properties of the stent and it retains its continuity even after deformation.

The amphilimus formulation consists of combination of sirolimus and long chain fatty acids. They are eluted together providing a combined effect. Sirolimus is a highly potent immunosuppressant agent with anti-proliferative, anti-inflammatory, and anti-microbial action. The fatty acids act as a non-polymer carrier modulating drug elution, bioavailability, and homogeneous drug distribution across the arterial wall through their permeability. Fatty acids play an essential role in the metabolic cell process, especially in diabetic patients. In a non-diabetic cell, adenosinetriphosphate (ATP) is generated through glucose and fatty acid pathway, in 30% and 70% respectively [30]. In a diabetic cell, as glucose metabolism is impaired, fatty acid uptake is increased, and fatty acids constitute the main source of ATP production [30].

The release of the antiproliferative agent together with its polymer free carrier is succeeded through the abluminal reservoir technology (Figure 3). The drug is loaded onto the stent platform in reduced surface areas where it is protected, permitting retarded controlled elution to the arterial wall up to 90 days post-implantation [31,32].

## 4. Clinical Application in CLI Patients

We conducted a literature search for studies reporting the use of non-polymer amphilimus drug eluting stents for infrapopliteal arterial disease in patients with CLI. The key words used in literature search were: Critical limb ischemia; Below the knee; Drug eluting stents; Cre8; Polymer free; Endovascular. We searched electronic databases (PubMed, MEDLINE, Google Scholar, and Cochrane) for relevant studies published up to 30 September 2022. Moreover, we searched World Health Organization, European Union Drug Regulating Authorities Clinical Trials Database, and ClinicalTrials.gov websites and scientific programmes of recent vascular and interventional radiology conferences for relevant studies.

Initial search of electronic databases, organization websites and registers by using the key words identified 343 articles. After excluding duplicates and irrelevant records by reading the title and the abstract we retrieved 216 relevant articles. From these articles, 38 studies mentioned the use of non-polymer drug eluting stents. Further, 35 studies were rejected because did not mention the use of the stents in CLI patients. The remaining three studies were included in our review (Figure 4). The first study (S1) was retrieved from PubMed electronic database: Tigkiropoulos et al. in their prospective single-center cohort study present the use of Cre8 stent in 27 patients with infrapopliteal arterial disease comparing it with a control group of 27 patients subjected to angioplasty with paclitaxel-coated balloons [33]. The second study (S2) was retrieved from Google Scholar electronic database: Santos et al. in their prospective study present the use of Cre8 stent in 10 patients with infrapopliteal arterial disease [34]. The third study (S3) was retrieved from LINC2022 conference: Sirvent presents on his oral presentation the use of Cre8 in 42 patients with infrapopliteal arterial disease [35]. Kaplan–Meier curve diagrams provided in S3 were used to extract numerical data about primary and secondary patency, limb salvage, freedom from clinically driven target lesion revascularization (CD-TLR), and mortality at 12 and 24 months. Main outcomes analyzed in our present review were: Primary and secondary patency, major amputation rate, freedom from CD-TLR and mortality reported in three above mentioned studies. Any other additional information would be provided by contact with the corresponding authors.

Totally 79 patients were included in all studies (72.2% were male, mean age 74 ± 3.8 years). Most of the patients (*n* = 65) were Rutherford class 5–6 (82.3%), diabetes mellitus (DM) was present in 66 patients (83.5%) and hypertension in 72 patients (91.1%). Baseline patient characteristics of all studies are presented in Table 2. One-hundred seven target lesions were treated with Cre8 stent (77.2% of patients received one stent, 19% two stents and 3.8% three to four stents). Occlusive infrapopliteal target lesion was present in 55% of patients in S1 and 52.4% in S3. S2 did not report the proportion of occlusive disease. The most common artery treated was tibioperoneal trunk. Mean stented length was 20 mm (15–40 mm) in S1 and 35 mm (30–60 mm) in S3 (not reported in S2). Mean primary patency of all three studies was 82.5% at 12 months. Primary patency of S1, S2, and S3 studies was 81%, 80%, and 83.8% respectively. S3 reported primary patency 83.8% at 24 months. Secondary patency was 96% at 12 months in S1 and 97.5% in S3 at 12 and 24 months. Secondary patency was not reported in S2. Mean limb salvage of all three studies was 91.3% at 12 months. S1, S2 and S3 studies have limb salvage rate 85%, 90% and 95.6% at respectively. S3 reported limb salvage 95.6% at 24 months. Freedom from CD-TLR in S1 and S3 was 96% at 12 months and 83.8% at 24 months respectively. S2 did not report respective findings regarding CD-TLR. Mortality was 15% in S1 study and 23.8% in S3 study [33,35], whilst mortality rate was not reported in S2. Table 3 summarizes the results of all included studies.

## 5. Discussion

This paper was performed to provide an overview of the novel polymer free amphilimus drug eluting stents Cre8 (Alvimedica, Instabul, Turkey) describing its special characteristics, the mechanism of action as well as its early clinical results in patients with CLI. In the therapeutic field of coronary artery disease, the role of amphilimus formulation as an antirestenotic agent has been well established in RCTs and large prospective studies [36,37,38,39]. In contrast, for peripheral arterial disease, the clinical data are scarce with three available studies evaluating safety and efficacy of the device since Cre8 DES has recently been introduced for BTK atherosclerotic disease [33,34,35]. All studies showed acceptable rates of primary patency, limb salvage, and CD-TLR at 12 and 24 months. To the best of our knowledge, these are the only studies available in the literature reporting the use of non-polymer amphilimus drug eluting stents in CLI patients. It should be mentioned that these studies have important limitations. They lack randomization, and they are single center studies with a small number of patients. An additional limitation is Sirvent’s report of possible conflicts of interest on his study presentation regarding Alvimedica company.

Cre8 DES has unique characteristics. The polymer free nature of the scaffold, the thin stent struts, and the presence of BIS may induce early endothelization of the arterial wall and may inhibit early in stent thrombosis. The Demonstrate study was a multicenter, randomized parallel group study, with a small number of patients, where Cre8 was compared Vision/Multilink8 BMS [40]. The primary endpoint was the ratio of uncovered to total stent struts per cross section (RUTTS) score of <30%, determined by OCT at 3 and 1 months for Cre8 and Vision/Multilink8 respectively. Secondary endpoints were the percentage of uncovered/malapposed stent struts, neointimal growth and thickness. RUTTS score < 30% occurred in 99.8% (899/901) of Cre8 struts and in 99.6% (1116/1121) of Vision/Multilink8 struts (difference 0.2, CI 95% −0.2 to 0.6, p for noninferiority < 0.001). The percentage of uncovered/malapposed struts was comparable (0.36 ± 0.64 vs. 0.12 ± 0.24, *p* = 0.145) in the two study groups, while both neointimal percentage area (8.46 ± 5.29 vs. 19.84 ± 15.93, *p* < 0.001) and thickness (0.07 ± 0.04 vs. 0.16 ± 0.12, *p* < 0.001) were significantly reduced by Cre8 DES. Authors concluded that Cre8 at three months has comparable strut coverage to Vision/Multilink8 BMS at one month and reduced neointimal hyperplasia formation rate.

The presence of long chain fatty acids as an amphilimus carrier is considered an important advantage of Cre8, especially in diabetic patients. It permits enhanced delivery, homogeneous distribution, and drug uptake in diabetic cells in contrast to other polymer DES. The vast majority of CLI patients are diabetics and Cre8 could gain ground for clinical use in the setting of diabetes in peripheral arterial disease. The first study that evaluated the efficacy of amphilimus against everolimus eluting stents in patients with CAD and diabetes was the Reservoir trial [38]. It was a multicenter, randomized, noninferiority trial where the primary endpoint was the neointimal volume obstruction assessed by optical coherence tomography at nine-month follow-up. The volume obstruction was 11.97 ± 5.94% for amphilimus versus 16.11 ± 18.18% for everolimus, meeting the noninferiority criteria (*p* = 0.0003). Authors concluded that Cre8 could provide an important advantage in diabetic patients undergoing percutaneous coronary angioplasty. SUGAR trial was a randomized non-inferiority study that evaluated the efficacy of amphilimus DES against zotarolimus eluting stents in patients with diabetes and coronary artery disease [41]. The main endpoint was target lesion failure, defined as a composite of cardiac death, target-vessel myocardial infarction, and CD-TLR at one year. One hundred six events occurred, 42 (7.2%) in the amphilimus group and 64 (10.9%) in the zotarolimus group (hazard ratio (HR): 0.65, 95% confidence interval (CI): 0.44–0.96; P non-inferiority < 0.001).

A potential relationship of the anti-proliferative drug eluting technology with mortality and major limb amputation was suggested in patients with PAD causing special concerns regarding their application. The overall safety and efficacy of the paclitaxel coated balloon angioplasty of infrapopliteal arteries was evaluated in a systematic review and meta-analysis of all the currently available industry randomized trials, by Katsanos et al. [42]. Patients with paclitaxel-based therapies have higher risk of death or major amputation compared to patients who underwent plain balloon angioplasty (13.7% vs. 9.4%, hazard ratio 1.52; 95% confidence interval: 1.12–2.07, *p* = 0.008). The current data regarding the safety and relationship of -limus eluting technology with mortality and major amputation are still scarce and under investigation due to the fact that results are limited because of small, non-randomized studies.

The efficacy of other -limus (sirolimus, everolimus) eluting stents for the treatment of infrapopliteal arterial disease has been analyzed in systematic reviews, prospective and retrospective cohort studies [43,44,45,46,47]. In a meta-analysis by Fong [43], individual patient data meta-analysis of 282 patients who underwent infrapopliteal stenting with sirolimus eluting stents yielded primary patency rates of 95.2% and 82.8% at 6 and 12 months respectively, whereas pooled six-month primary patency (339 patients) was 87.3%. A prospective study by Giaquinta et al. evaluated the efficacy of polymer coated everolimus eluting stents (Xience Prime) in CLI patients [44]. The study included 122 patients, and 52.5% were diabetics, with mean lesion length 52.7 mm. The primary patency rate was 88.9% at 1-year. The survival, amputation-free survival, and freedom from TLR rates were 88.1%, 93%, and 91.5% at 1 year respectively. A retrospective real-world study by Aburahma et al. investigated the efficacy everolimus eluting stents (Xience; Abbott Vascular, Santa Clara, Calif). The results were not satisfactory and were inferior to previously reported outcomes with primary patency reaching 57% at one year [45]. The PREVENT study was a prospective, multicenter, non-randomized study that evaluated the everolimus eluting stent “Promus Element and Promus Element Plus stents” (Boston Scientific, Marlborough, MA, USA) [46]. Primary patency was 86.2% and freedom from TLR was 93% at 1 year. Table 4 summarizes the main outcomes of everolimus DES studies in CLI patients. Amphilimus DES studies have similar results rearding primary patency and CD-TLR at one year with most everolimus DES studies. Further studies should be conducted to evaluate efficacy between different -limus DES, especially in diabetic CLI patients.

## 6. Conclusions

This is the first review in the published literature on CLI patient outcomes with Cre8 DES. Treatment of BTK atherosclerotic lesions in patients with end-stage peripheral arterial disease and diabetes with the polymer free amphilimus eluting stent seems to be safe and efficacious with results similar or superior to other contemporary DES. A larger number of patients, especially diabetics, should be included in multicenter, prospective or randomized studies to evaluate this new stent technology and long-term clinical outcomes using Cre8 DES.

## Figures and Tables

**Figure 1 medicina-59-00039-f001:**
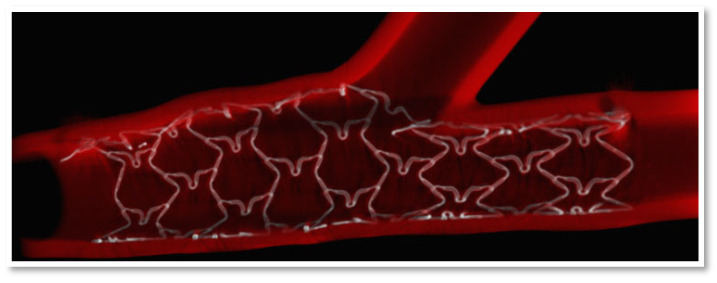
Optimal scaffolding of the Cre8 eluting stent in arterial bifurcation. With permission from Alvimedica brochure [accessed 18 October 2022] Retrieved from: http://www.alvimedica.com/.

**Figure 2 medicina-59-00039-f002:**
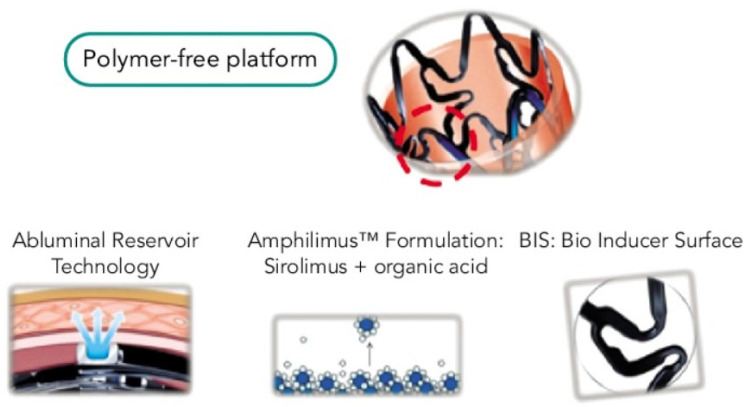
Features of Cre8 eluting stent. Polymer-free platform with abluminal reservoir technology as a drug delivery system for amphilimus distribution to the arterial wall and Bio Inducer Surface (BIS) with anti-thrombotic and healing properties [29].

**Figure 3 medicina-59-00039-f003:**
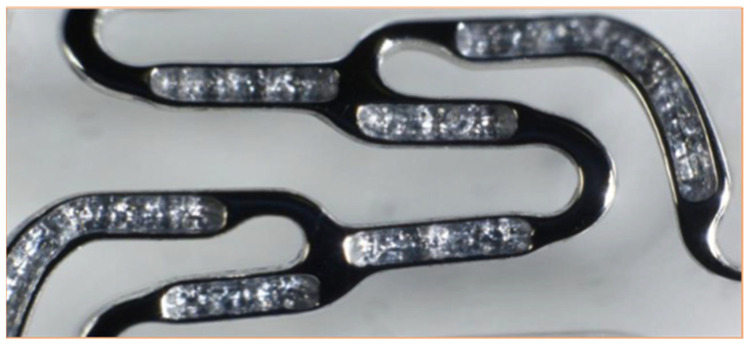
The abluminal reservoirs of Cre8 eluting stent [32].

**Figure 4 medicina-59-00039-f004:**
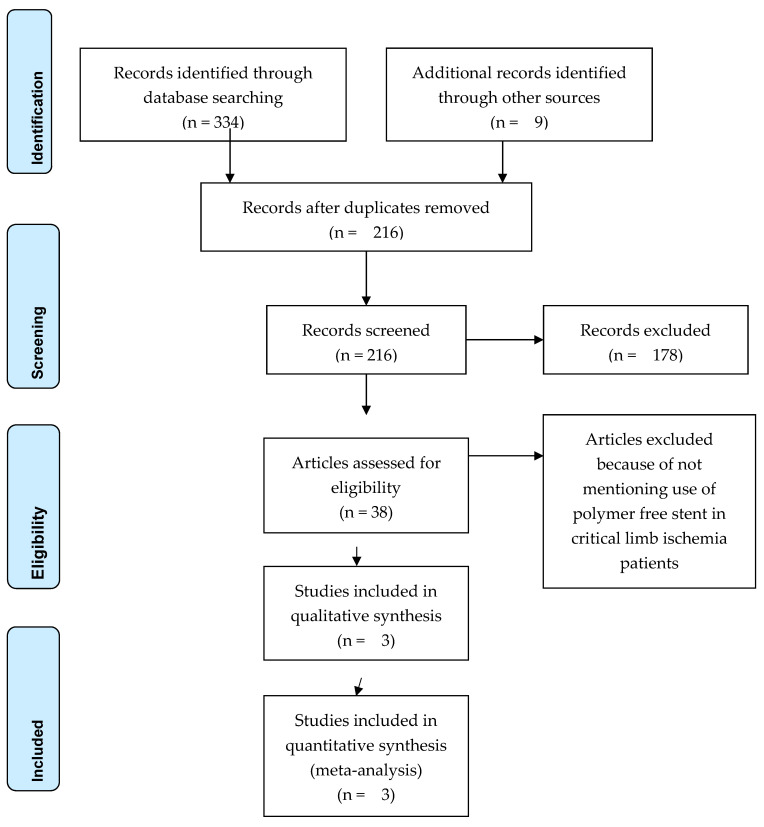
PRISMA flow chart showing study search and selection.

**Table 1 medicina-59-00039-t001:** Characteristics (number of cells, strut thickness) of Cre8 amphilimus eluting stent per stent diameter.

Nominal Diameter (mm)	Number of Cells per Circumference	Strut Thickness (Microns)
2.25	4 cells	70
2.5–2.75	4 cells	80
3.00–3.50	5 cells	80
4.00–4.50	6 cells	80

**Table 2 medicina-59-00039-t002:** Baseline patients’ characteristics in 3 studies.

Studies	Patients (*n* = 79)	Mean Age	R.C 5–6	Male	Hypertension	D.M	ESRD	DSL	CAD
Tigkiropoulos [33]	27	79	88.8%	74%	100%	100%	30%	11%	33%
Santos [34]	10	69	60%	40%	80%	80%	11%	N/R	N/R
Sirvent [35]	42	76	88.1%	78.6%	91%	73.8%	N/R	74%	N/R

Abbreviations: Rutherford Category (R.C), Diabetes Mellitus (D.M), End Stage Renal Disease (ESRD), Dyslipidemia (DSL), Coronary Artery Disease (CAD), Not Reported (N/R).

**Table 3 medicina-59-00039-t003:** Clinical outcomes of polymer free amphilimus eluting stent in 3 studies.

Studies	Lesions Treated (*n* = 107)	Occlusive Lesions (%)	Stent Length (mm)	Primary Patency	Secondary Patency	Limb Salvage	Mortality	Freedom from CD-TLR
Tigkiropoulos [33]	31	55%	20	81%	96%	85%	15%	96%
Santos [34]	10	NR	NR	80%	NR	90%	NR	NR
Sirvent [35]	66	52.4%	35	83.8%	97.5%	95.6%	23.8%	83.8%

Abbreviations: Clinically Driven Target Lesion Revascularization (CD-TLR), Not Reported (NR).

**Table 4 medicina-59-00039-t004:** Summary of primary endpoints of everolimus drug eluting stents of above studies.

Author	Study	Patients	Mean Lesion Length	Primary Patency at 1 Year	Freedom from TLR at 1 Year	Limb Salvage (No Major Amputation) at 1 Year
Giaquinta [44]	Prospective	122	52.7 mm	88.9%	91.5%	93%
AbuRahma [45]	Retrospective	98	41 mm	57%	92%	87%
Taeymans [46]	Prospective	70	22.83 mm	86.2%	93%	100%
Bosiers [47]	Prospective	60	47.4 mm	75.4%	84.9%	94.4%

Abbreviation: Target Lesion Revascularization (TLR).

## Data Availability

Electronic databases were used for extraction of data.

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
