# Peer review of "Polymer Free Amphilimus Drug Eluting Stent for Infrapopliteal Arterial Disease in Patients with Critical Limb Ischemia: A New Device in the Armamentarium"

_medicina, 2022, doi:10.3390/medicina59010039_

Round 1

Reviewer 1 Report

The authors report a review on studies investigating a polymer free, amphilimus drug eluting stent for infrapopliteal arterial disease in patients with chronic limb threatening ischemia (CLTI). As patency rates after endovascular below the knee (BTK) interventions are low and high restenosis rates are reported, novel technologies for BTK interventions are of high interest. Several issues should be considered regarding the manuscript:

Abstract: The average lesion length of the lesions in the included studies should be reported in the abstract.

Introduction:

As mentioned by the authors, prior studies on drug-eluting stents for BTK disease have shown favorable results. However, it should be highlighted that rather short, proximal lesions have been treated with these stents while many BTK lesions - especially in CLTI patients - are often long occlusions. Thus, stenting is only for a small proportion of lesions an option. This aspect should be discussed in the introduction and discussion sections.

Methods and Results:

Why was the search strategy limited to chronic limb threatening ischemia? Is there any data for non-CLTI patients available? 

A flow diagram should be presented for the results of the systematic literature search. How many titles were initially identified and then excluded based on the abstracts?

In addition to clinical outcomes as shown in table 2, please also present baseline study characteristics as a table.

Table 2: Please indicate the time of follow-up in the studies.

Discussion: 

The authors use both expressions CLI and CLTI throughout the manuscript. Please unify these expressions.

For comparison with prior literature in the field, please provide a table showing the outcomes of infrapopliteal stenting with other drug-eluting devices. In this way, the discussion can be modified by describing less detailed results (i.e. less numbers) of other studies/reviews in this area. At the moment, the readability of the discussion is limited by too many numbers.

Figures: 

Figure 5: The description of the y-axis unclear. Please specify to what cumulative % released is referring to.

Author Response

  1. It should be included in the abstract. Mean lesion stented length was 20 mm and 35 mm in 2 studies.
  2. Dear reviewer, I totally agree with you. PTA remains the primary therapy of infrapopliteal disease regardless the length of the lesions according to the recent guidelines of interventional radiology committee. Stenting is used mainly as a bailout technique after elastic recoil of the artery or local dissection that affects distal perfusion of the leg.We will emphasize in the text this anatomic characteristic of cli patients.
  3. Dear Reviewer, there is a scarce data regarding infrapopliteal stenting in patient with claudication.Usually these patients have aortoiliac of femoropopliteal disease with good tibial run off vessels that do not require infrapopliteal intervention. 
  4. You are absolutely right. A flow diagram will be included in our review with included and excluded studies.
  5. The baseline characteristics of patients in 3 studies will be analyzed in a separate table as well as the follow up period will be added.
  6. Dear reviewer you will use the term CLI and not CLTI in our manuscript.
  7.  A table will be added in the text with the primary outcome of other limus stents implanted in CLI patients from real world original not sponsored studies to evaluate their efficacy and safety.
  8. The y-axis in figure 5 represents the percentage of drug release during a period of time in experimental porcine arteries according to a experimental study evaluating pharmacokinetics of DES

Reviewer 2 Report

The paper describes a novel drug eluted stent and its applications on the treatment of below the knee arteries in patients complain with clinical limb ischemia. The device is described in detail, both in the morphology of the stent and in the particular drug released. 

Nevertheless I have the following doubts:

1)    The literature review is based on the analysis of two published studies and one oral presentation, given by a very quoted speaker during a prestigious international meeting. However, data are very scarce and fragmentary, it is really hard to obtain robust results.

2)    Among the reported results of the studies, two of them declare a mortality at 12 months of 15% and 23.8%. I guess Authors should explain this finding, considering that recently the overall mortality is central when a drug-eluted technology is applied. Moreover, those percentages do not appear aligned with other studies: Fong et al. (reference number 41 cited by the Authors) declare a mortality at 6 months of 5.4%; Giaquinta et al. (reference number 42 cited by the Authors) declare a mortality at 12 months of 11.9%; ACHILLES trial reports a mortality at 1 year of 10.1% and 11.9% in the two groups.

3)    Similarly, regarding primary patency, at page 6 Authors declare a mean primary patency of 82.5% at 12 months but only two of the three studies analyzed reported this data and in one study it is referred to 24 months.

4)    I notice probably a typo on page 4, first line of the section ‘Discussion’ (sorry but in the version of the paper downloaded lines are not numbered, so it is not simple to refer to specific words) where ‘sirolimus’ should be changed in ‘amphilimus’.

The description of the novel device is complete e precise, deepening into molecular specifications of the drug and its potential advantages on CLI patients. The second part, concerning data available to date and its interpretation seems a bit scarce: for example, last paragraph of the section ‘Discussion’ ends with a series of data without comments. I would appreciate a more detailed data reporting (e.g. with a table of the results as patients’ and lesions’ characteristics) and a more articulate discussion, focusing on the potential advantages of the novel device (e.g. In diabetic patients). On the other hand the first part, with table, graph and images obtained from the commercial brochure appears somewhat lengthy.

Author Response

1.Dear Reviewer,I totally agree with you comment.There is a scarce of data regarding efficacy of Cre8 in CLI patients. All 3 studies have  limitations such as the small number of patients and the lack of randomization.However it is the first review of Cre8 in infrapopliteal disease and it should be considered as a first step describing its application in peripheral arterial disease. Drug eluting technology is continuously evolved and results should be published even with small number of patients.

2. As it concerns the mortality of CLI patients, I would like you to inform you that mortality in CLI patients ranges up to 25% at 1 year.In real world studies, mortality numbers are really controversial.At observational study by Steunenberg (doi:10.1177/153857441985478) 1 year mortality in endovascular group was 40%. In another study by Biasi (doi: 10.23736/S0021-9509.16.09159-x) overall survival by Kaplan-Meier was 68%  at 1 year. In an observational study by Roijers 12 month mortality was 30% ( DOI: 10.1016/j.jvs.2019.08.245).As we can all understand mortality ranges from less than 10% to 40%. Etiology is multifactorial  depending on the region and comorbidities of patients.

3. Dear Reviewer, we mentioned in our manuscript in the section of "Clinical application in CLI patients" that results at 12 months were extracted from Kaplan Meir curves regarding primary end points at 12-24 months in Sirvent study.

4. The word Sirolimus will be changed with amphilimus.

5. Dear Reviewer, we will add in the discussion a table with outcomes of other limus DES in CLI patient from observational studies, so we can see efficacy of Cre8 in relation to other DES.Cre8 has been evaluated in diabetic patients with coronary artery disease and not in CLI patients. It could be possible to add these results to improve our paper. 

I would like to thank you for your comments.

Round 2

Reviewer 2 Report

Dear Authors,

I appreciate the revision made to your paper. I still think that the first part is somehow lengthy, but it could be appropriate for publication on this journal.

Regarding the Authors' notes:

1. In case of very scarce data I do not believe that the calculated averages of the results have sense, I would prefer to report the raw data as they are published on the single studies, adding appropriate comments on the 'Discussion' section.

2. Thank for underlining the mortality rates of CLI, which is a very life-threatening endemic burden and is really essential to maintain the attention to this fact. However, probably I did not explain well that I intend to focus on the mortality rates after the application of a drug eluted technology, and not on the general mortality among CLI patients. Drug eluted balloons and stents are been blocked by a controversial metanalysis in the very recent years for a suspection in a major mortality rate: I think it is an issue to be discussed when an anti-proliferative drug is applied.

3. I do not believe that data extrapolated froma a kaplan-meir curve are strong enough to be used for other statistical analisyese, but this is just a personal opinion.

4. Thank you for the correction.

5. The table you kindly added are useful, and the general structure of the final part of the 'Discussion' section are now better

I would just recommend to revise the numeration of the table, even in the text, as now there are two 'Table 1'.

Author Response

Dear Reviewer,

I would like to thank you for your comments regarding our manuscript.

  1. We have added in the results  the primary patency and limb salvage rate separately for each study apart from the mean value. Abstract is already 300 words and we can't exceed it.
  2. We have added a paragraph regarding the relationship of drug eluting technology with mortality and amputation rate in Paclitaxel and -Limus according to the current bibliography.
  3. Thank you very much for your comment. We believe that Kaplan-Meier is a reliable statistical method in order to retrieve some results during follow up of patients, even when the number of patients is small.Don't forget that  studies regarding limbs eluting stents in CLI patients are limited.
  4. We have corrected table 1 in the text.